# STFANet: A spatial and temporal feature aggregation network for fake face detection in videos

Guoren Yao[1¤]*, Gaoming Yang[2], Xintian Liu[3], Lei Chen[1]

1 School of Computer Science, Huainan Normal University, Huainan, Anhui, China, 2 The First Affiliated Hospital of Anhui University of Science and Technology (Huainan First People's Hospital), Huainan, Anhui, China, 3 College of Electrical and Automation Engineering, Hefei University of Technology, Hefei, Anhui, China

¤ Current address: School of Computer Science, Huainan Normal University, Huainan, Anhui, China
* yaogr56@163.com

## Abstract

The verification of video authenticity has become progressively more challenging with the rapid advancements in video synthesis technologies. However, current detection approaches predominantly depend on intra-frame spatial artifacts and temporal inconsistencies, restricting their capacity to fully exploit the spatio-temporal characteristics of manipulated videos. To address this problem, we propose the **S**patial and **T**emporal **F**eature **A**ggregation **N**etwork (STFANet), which employs a two-path structure to extract spatial and temporal features independently. These extracted features are subsequently integrated to construct high-fidelity spatio-temporal representations. Additionally, we incorporate a Vision Transformer module to capture global dependencies within the feature maps, enhancing the overall feature representation. Extensive experiments validate the efficacy of the proposed approach in detecting facial forgery in videos. Performance evaluations on benchmark datasets, including Face-Forensics++ and Celeb-DF, confirm the effectiveness of our method, yielding AUC scores of 0.9933 and 0.9829, respectively. Furthermore, we investigate the impact of feature aggregation at different stages on the generated feature maps, revealing significant improvements in the quality of spatio-temporal representations.

## 1 Introduction

The human face has been regarded as a reliable and authentic identity marker [1]. However, with rapid advancements in deep learning and the available of computing resources, sophisticated generative models can easily manipulate face images and videos. These manipulations often involve altering facial attributes or even changing identities, making it challenging to distinguish individuals [2]. Consequently, the credibility of personally identifiable information is danger. While facial forgery techniques have found legitimate uses in education, video rendering [3], and image restoration

**Data availability statement:** All data used in this study are from publicly available sources, including the FaceForensics++ dataset (https://github.com/ondyari/FaceForensics). The experimental code and additional supporting files are available on the GitHub repository at https://github.com/yaogr56/STFANet/tree/master.

**Funding:** The author(s) received no specific funding for this work.

**Competing interests:** The authors have declared that no competing interests exist.

to promote diversity in entertainment and culture [4], their malicious application by unscrupulous individuals poses a threat to personal reputations and undermines public trust in media [5]. Therefore, the detection of forged faces is crucial for preserving trust for and safeguarding individual identities.

Facial deep forgery, commonly referred to as Deepfake, predominantly occurs in both image and video [6]. In image-based forgery, Autoencoders [7] or Generative Adversarial Networks (GANs) [8] are typically trained on both original and target faces to generate highly realistic forgeries. These methods are widely employed for identity swapping and face reenactment. Identity swapping involves replacing the source face with the target face, and face reenactment focuses on modifying facial attributes and replicating the pose, expression, and other characteristics of the target face.

Due to advancements in forgery and sophisticated post-processing operations, the forged images exhibit minimal visible artifacts, making the detection a challenge. In video-based facial forgery, maintaining temporal consistency across frames is critical. However, most tampered video frames are manipulated independently resulting in distinct forgery traces within each frame. These intra-frame forgery traces indirectly lead to visually inconsistencies between frames, such as flickering lights and facial position jitter, which disrupts the temporal coherence of the video.

Based on these findings, researchers have concentrated on investigating inconsistencies within facial regions across video frames and inter-frame correlations to improve the detection of face forged videos. The differences in intrinsic patterns between forged and original video frames in terms of RGB information [9], facial texture [10], and frequency domain features [11] have significantly influenced the advancement of face forgery detection techniques. However, the effectiveness of video frame analysis is highly dependent on the dataset used, and the detection performance may degrade significantly when applied to unseen datasets. As a result, ongoing research aims to develop more robust detection methods that integrate both intra-frame inconsistencies and inter-frame correlations.

Optical flow [12] and Recurrent Neural Networks (RNN) [13] are frequently utilized to model temporal correlation in video sequences. For instance, Amerini et al. [14] investigated motion anomalies between a person and the background by leveraging optical flow. They then integrated this inter-frame anomaly feature into a CNN model for identifying forged videos. Based on this work, Yang et al. [15] improved the modeling of temporal information by incorporating Long Short-Term Memory (LSTM) into the detection framework. However, it is important to note that the optical flow approach relies on the assumption of constant luminance, which may not hold in the case of fake videos. Faked videos often undergo luminance adjustments to enhance their realism, thereby limiting the applicability of optical flow methods in these scenarios.

To mitigate this limitation, Güera et al. [16] introduced the Inception-LSTM architecture, integrating the Inception module for intra-frame feature extraction with LSTM for temporal modeling. This design sought to leverage the complementary strengths of both modules to improve fake video detection. Sabir et al. [17] introduced the Bi-RNN architecture, replacing the LSTM with Bidirectional Recurrent Neural

Networks (Bi-RNN). This modification improved the temporal modeling capability, leading to enhanced performance. However, these RNN-based methods employ CNNs for extracting inter-frame features before feeding them into the LSTM or Bi-RNN. While CNNs excel at capturing spatial information, they struggle to handle temporal correlation, leading to the loss of important temporal information in the detection process. In contrast, 3D Convolutional Neural Networks (3D CNNs) [18] have the advantage of considering both spatial and temporal information within video frames, enabling them to learn spatio-temporal features. However, the limited perceptual field of the 3D convolutional kernel may hinder the effective analysis of complex spatio-temporal features. Additionally, the spatio-temporal feature sequences obtained through 3D convolutions often lack long-term dependencies, which can limit their ability to capture nuanced temporal patterns in video forgery.

Although existing research has made significant progress, there are still problems such as focusing only on the spatial features of a single frame or the temporal consistency between video frames and lacking comprehensive utilization of spatiotemporal features. There are even some methods that rely on specific types of forgery techniques and have limited generalization capabilities. To address these problems, this paper proposes STFANet, consisting of two modules, to obtain more representative spatio-temporal features of fake videos. In the first module, we focus on fully the spatial and temporal features by a two-path structure with different receptive fields. This allows us to capture both fine-grained details and motion-related features effectively. Additionally, feature aggregation is applied after each extraction stage to strengthen inter-feature correlations and refine feature quality. In the second module, we introduce the Vision Transformer (ViT) to achieve feature interaction on the extracted feature sequences. By incorporating ViT, we enhance the long-term dependency of the feature sequences, enabling a more comprehensive understanding of the temporal dynamics in the video. This module also facilitates obtaining a global feature representation of the synthetic video.

The STFANet model can effectively integrate spatiotemporal features, and achieve robust detection of different types of deep fake videos through global feature modeling and generalization capability improvement. The main contributions of this paper are as follows:

- We propose a method based on Spatial and Temporal Feature Aggregation Network (STFANet) for face video forgery detection, which obtains a comprehensive representation of the forged video content, thus avoiding the inherent limitations of using a single spatial or temporal feature representation to detect forged videos.
- With STFANet, we can sample video frames at different time steps to extract spatially and temporally inconsistent features. Additionally, we combine stage-by-stage feature aggregation and integrate the Vision Transformer module to improve the quality of the obtained spatiotemporal features.
- Extensive experiments were conducted to evaluate STFANet, and results from both intra-dataset and cross-dataset evaluations confirm its effectiveness on well-known datasets, demonstrating its robustness in face forgery detection. Furthermore, our method exhibits generalizability to unknown datasets, as evidenced by the results of inter-dataset experiments.

This structure of this paper is as follows. Sect 2 provides an overview of the research progress made in detecting forged face videos. Sects 3 and 4 present the working principle of our proposed model, along with the experimental details, results, and ablation studies. Finally, Sect 5 concludes the paper and discusses potential avenues for future research in the field of video face forgery detection.

## 2 Related work

### 2.1 Detection of image-level forged face video

The generation of fake face videos typically involves the fusion of a synthesized face with an existing background on a frame-by-frame basis. This process introduces specific artifacts that result in subtle differences in the spatial and frequency domains. To detect these forged features, researchers have developed robust detectors that leverage various

spatial artifacts, including image texture, face landmarks, blending boundaries, and more. In general, the original face region usually contains more details compared to the forged face, which leads to subtle differences in RGB statistical information between real and forged frames.

To exploit these differences, Xia et al. [19] proposed the construction of three types of texture maps: pixel-level maps, block-level maps, and region-level maps. By amplifying the differences in color components through these texture maps, the authors achieved good classification results with a simple training approach. Wang et al. [20] employed feature detection and description algorithms to demonstrate that real faces possess more discriminative features compared to fake ones. In another approach, Li et al. [21] focused on analyzing the foreground and background of falsified face images, leading to the identification of a distinct hybrid boundary referred to as the face X-ray. This boundary serves as a distinguishing characteristic between genuine and manipulated face images. On the other hand, Qian et al. [22] concentrated on deep forgery features in the frequency domain. They proposed F3-Net, a method that extracts two frequency domain features to effectively detect low-resolution forged image videos. Luo et al. [10] extracted multi-scale high-frequency features and low-level image texture features to enhance the generalization performance of forgery detection. By incorporating information from multiple scales and considering image texture features, their approach achieved improved performance across various forgery detection tasks. Xia et al. [23] proposed an Inspector framework to defend face forgery detectors against adversarial attacks. Without modifying the detector architecture or preprocessing inputs, Inspector performs a coarse-to-fine defense. Bai et al. [24] analyzed the root cause of the detector's insufficient cross-domain performance from the perspective of causal learning, pointing out that the model is prone to learning spurious correlations between semantic features and labels that are irrelevant to forgery. They thus proposed the Feature Independence Constrainer (FIC) and Feature Alignment Module (FAM), which establish high-order correlations between the spatial and frequency domains to extract more discriminative forgery clues, and combine fine-grained high-frequency feature extraction to enhance forgery artifact learning.

While image-level forged face video detection methods have shown success in revealing discriminative features present in forged frames, these methods face two primary challenges. Firstly, detection algorithms are typically applied independently to each video frame, and the average fraction of frames is used for video prediction, which can result in sensitivity to individual frame prediction results and may not fully capture the overall video-level characteristics. Secondly, these methods often do not fully exploit the specific features of forged face videos, and the lack of modeling inter-frame artifacts can limit their detection performance. In contrast, our proposed method addresses these issues by performing direct video-level prediction, which leads to more reliable results. We take into account both spatial information and temporal features, allowing for comprehensive spatio-temporal feature extraction that aids in classification. By considering the entire video sequence as a whole, our method captures the inter-frame artifacts and exploits the temporal dynamics of the forgery, leading to enhanced detection performance.

## 2.2 Detection of video-level forged face video

In the context of forged videos, the manipulation disrupts the natural motion continuity of faces and introduces temporal tampering traces, which can be utilized for recognition and detection purposes. Many methods have leveraged these temporal anomalies to detect video forgeries. Biological information related to the face, such as blink activity [25], head deflection [26], gaze direction [27], and lip movement [28], has been utilized to extract abnormal motion features from face videos. However, these methods often focus on specific regions of the face, potentially leading to the loss of the complete feature representation. To address this issue, some researchers have proposed approaches that input the entire facial region of video frames into the model, aiming to capture a comprehensive representation of the face. For example, Sabir et al. [17] input the complete frame to a CNN model and model the obtained feature sequences using RNN to capture the temporal information. Another approach, proposed by Zheng et al. [29], is an end-to-end temporal feature

extraction framework. This framework enhances the representation of temporal features by suppressing the extraction of spatial features from the model.

Based on the work, Hu et al. [30] designed a two-stream network that combines frame-level spatial information and video-level temporal information for feature extraction. To enhance the representation of inconsistent features, Mehra et al. [31] proposed motion amplification, utilizing 3D residuals to extract spatio-temporal features in videos. Pu et al. [32] combined CNN with RNN to extract features and model temporal dependencies. They obtained a feature map where frame-level and video-level predictions were performed simultaneously To address the challenge of imbalanced real-world fake and real video data, they introduced a joint loss function that enables better learning and more effective model training. Furthermore, Wu et al. [33] introduced the SSTNet method, which integrates spatial, temporal, and steganographic analysis features to enhance the generalization and robustness of the forgery detection model. By considering multiple aspects of the video content, their approach improves the model's ability to detect various types of forgeries. In addition, Multiple Instance Learning provides a new perspective for video face forgery detection. Li and Lang et al. [34] treated the face video frames as instances and the video as a bag, enabling direct video-level prediction. To address the limitations of previous datasets that often lack video data with mixed real and fake frames, they introduced a new dataset called FFPMS specifically designed for face forgery detection at different levels of granularity.

Although various methods have been proposed for fake face detection by spatio-temporal features, these approaches still face two primary challenges. Firstly, existing detection methods often compromise the representation of intra-frame spatial features when focusing on modeling temporal information. This trade-off can limit the effectiveness of capturing fine-grained spatial details in forged face videos. Secondly, the extracted spatial and temporal features are typically treated independently, lacking proper correlation in the final spatio-temporal feature representation. This will result in a suboptimal integration of both spatial and temporal cues. In contrast, our proposed model addresses these issues by maintaining the preservation of frame-level spatial features while simultaneously extracting temporal features through feature aggregation operations. By incorporating feature aggregation, we can effectively fuse the spatial and temporal information to create a more comprehensive spatio-temporal feature representation. Furthermore, our model incorporates a Vision Transformer, which facilitates interaction with the spatio-temporal feature sequences.

## 3 The proposed STFANet

This section presents the proposed classification model, STFANet, which combines 3D CNN networks and ViT, to effectively identify real or face fake videos generated by different forgery algorithms. The model follows a process that involves extracting frames from the video, performing face cropping, and generating video sequences for detection. The objective of is to leverage the spatial artifacts present in manipulated video frames and the temporal inconsistencies across frames to achieve accurate classification and capture representative patterns of forged videos. To accomplish this, we introduce a two-level path module. This component allows us to separately capture the temporal and spatial features of synthesized videos. By integrating these two types of features, we can obtain strongly correlated spatio-temporal feature sequences. These sequences are then input into the ViT module, which enables the extraction of global spatio-temporal features. The working principle of our model is illustrated in Fig 1, highlighting the integration of spatial and temporal information. We will explain in detail the implementation of each stage of the detection method in the following sections

### 3.1 Data preprocessing

In this stage, the video is processed by decomposing it into individual frames. As our focus is on the facial region, it becomes necessary to crop the face from each frame. To accomplish this, we utilize MTCNN (Multi-Task Convolutional Neural Network), a lightweight and highly accurate face detection model. MTCNN enables us to identify the bounding

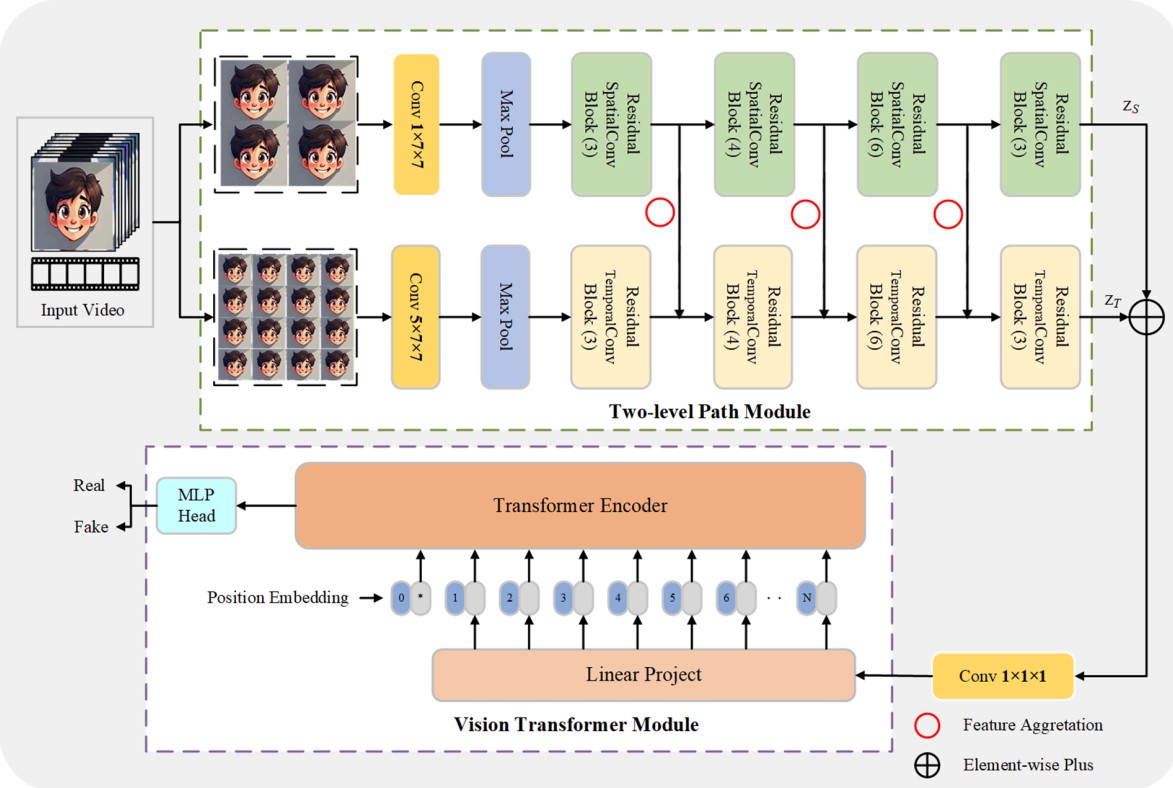

**Fig 1. The working principle of the proposed STFANet.**

boxes of potential face locations within each frame. Once the face regions are identified, they are extracted for further processing. To maintain context and differentiate the cropped face image from its surroundings in the database, we retain a narrow edge region with a width of 50 pixels. This edge region captures some background information surrounding the face. The resulting face images are saved with a size scaled to the default input resolution of $224 \times 224$ pixels, which is suitable for our model. This process is applied to each frame, which generates a video sequence composed of these face images.

## 3.2 Tow-level path module

The first component of our model employs a two-path structure to learn spatial semantic features and temporal information across video frames, which enables the simultaneous extraction of spatial features and temporal features. Through the process of stage-by-stage feature aggregation, we are able to obtain spatio-temporal feature sequences.

**3.2.1 Spatio-temporal feature representation.** We employ a two-path structure as the fundamental framework of our model. The first path serves as a spatial feature extractor, responsible for extracting spatial features within individual video frames. The second path acts as a temporal feature extractor, capturing temporally inconsistent features across multiple frames. Both paths receive inputs from the same video sequence, which is continuously sampled at different temporal intervals. An $N$-frames video sequence $V = p_{i=1}^{i=N}$ can be divided into subsets containing different numbers of frames. The sampling steps of the spatial and temporal paths are denoted by $u$ and $v(u>v)$, respectively. Thus, the input of the spatial path is $S = \{pu, p_{2u}, \cdots, p_N\}$, and the other input of the temporal path is $T = \{p_v, p_{2v}, \cdots, p_N\}$. No specific requirements for the network structure are necessary for the two paths used for feature extraction, and standard backbone networks can

be employed. So we utilize ResNet50 as the backbone network for feature extraction in both paths in our experimental setup. In particular, all bottleneck blocks are modified to focus more on specific tasks in each path. As depicted in Fig 2, we introduce our proposed Residual SpatialConv Block(RSCB) and Residual TemporalConv Block(RTCB). These blocks are utilized to replace the original bottleneck blocks in both paths.

The spatial path utilizes RSCB to obtain an input tensor containing $N/u$ frames, employing a large time sampling interval $u$. More feature maps are computed to capture richer information. 3D CNN models typically use convolutional kernels $K_t \times K_h \times K_w$, where $K_t$, $K_h$, and $K_w$ represent the temporal, height, and width dimensions, respectively. These kernels are usually larger than 1. A special design is adopted for the feature extraction to focus on spatially relevant features, as depicted in Fig 2(a). That is, we set the temporal dimension of the convolutional kernels used in RSCB to 1. This involves replacing the original kernels with $1 \times K_h \times K_w$ kernels, which limits their ability to handle temporally relevant features. The convolutional setup can be seen as a spatial convolutional filter with 2D, enabling it to effectively learn discriminative features. In contrast, the extraction of temporal features is performed through a separate path. By computing a reduced number of feature maps, the spatial modeling capacity is reduced, allowing for a stronger emphasis on capturing significant temporal information between video frames. To specifically target the extraction of temporal information while preserving the location information of the input features, we introduce the Residual TemporalConv Block (RTCB) as depicted in Fig 2(b).

**3.2.2 Feature aggregation.** To improve the correlation between features, we incorporate feature aggregation, as illustrated in Fig 3, after the feature extraction. In our model, we designed a two-level path module for feature extraction, utilizing the well-established ResNet50 architecture. The feature representation at any layer during the feature extraction

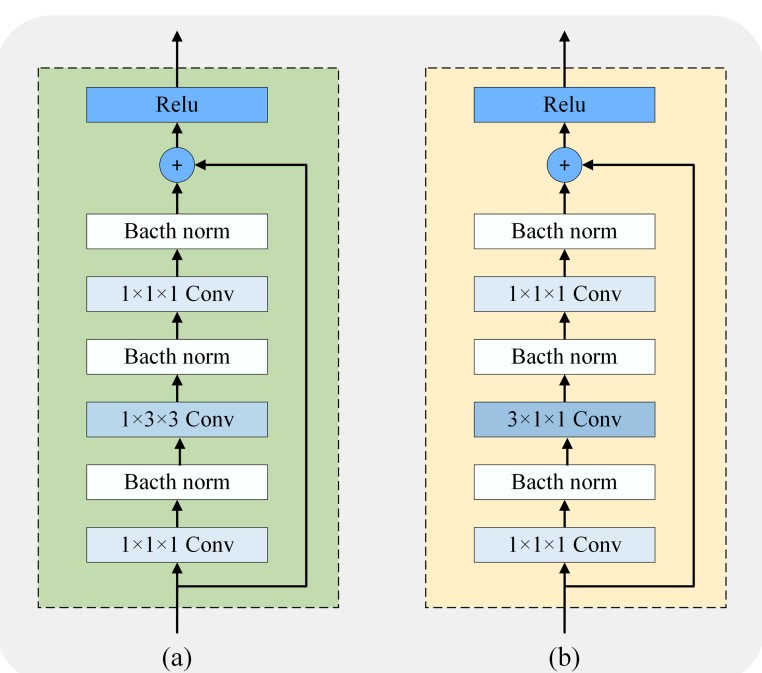

**Fig 2. The structure of the Conv Block.** (a) Residual SpatialConv Block. (b) Residual TemporalConv Block.

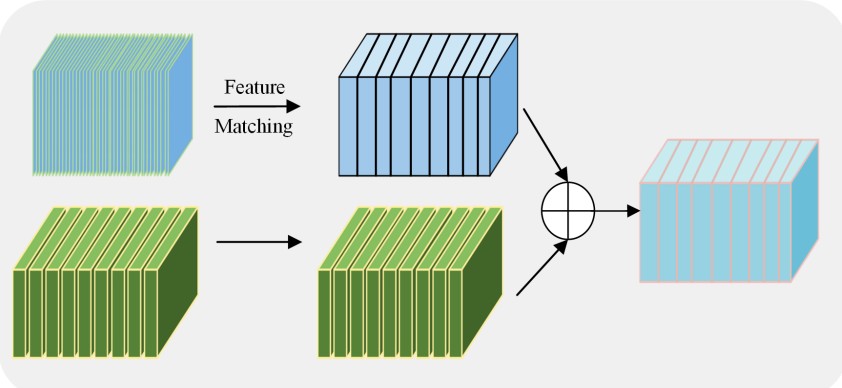

**Fig 3**. **The pipeline of feature aggregation.**

process is defined as follows:

$$\mathbf{z}_l = \mathbf{z}_0 + \sum_{i=0}^{l-1} F(\mathbf{z}_i, \mathbf{W}_i), \tag{1}$$

where $l$ is an arbitrary layer of a network, $\mathbf{z}_0$ is the feature map of the initial input, and $F()$ is the residual function. According to formula (1), we describe the feature map of any layer of the $s$ and spatial path $t$, respectively, as follows:

$$\mathbf{s}_l = \mathbf{s}_0 + \sum_{i=0}^{l-1} F(\mathbf{s}_i, \mathbf{W}_i), \tag{2}$$

$$\mathbf{t}_l = \mathbf{t}_0 + \sum_{i=0}^{l-1} F(\mathbf{t}_i, \mathbf{W}_i), \tag{3}$$

Feature aggregation is performed to facilitate interaction between temporal and spatial features. Through the feature extraction process, we obtain temporal and spatial feature maps with different shapes. The feature shape of input to the temporal path is $\{T, W, H, C\}$, while the feature dimensions of input to the spatial path are $\{mT, W, H, nC\}(m \times n = 1)$. $T$ denotes the number of input frames; $W, H$, and $C$ denote the width, height, and number of channels of a single video frame, respectively. To achieve feature matching, the temporal dimension of the feature maps in the spatial path is converted into channels. So, we can reshape $\{mT, W, H, nC\}$ into the shape of $\{T, W, H, mnC\}$. After feature matching, we obtain spatial path features with the same shape as the temporal path feature maps. Further, the feature map of the spatial path was added to the feature map in the temporal path. Thus, the temporal feature map at the corresponding aggregation stage is formalized as follows:

$$\mathbf{t}_l = \mathbf{t}_0 + \sum_{i=0}^{l-1} F(\mathbf{t}_i, \mathbf{W}_i) + \mathbf{s}_l. \tag{4}$$

The final spatio-temporal feature is as follow:

$$\mathbf{X} = \mathbf{z}_s + \mathbf{z}_T, \tag{5}$$

where $\mathbf{z}_S$ and $\mathbf{z}_T$ are the extracted spatial and temporal features, respectively.

                                                    

## 3.3 Vision transformer module

This module focuses on the interrelationships of the acquired spatio-temporal feature sequences. The first module produces the spatio-temporal feature representation $\mathbf{X} \in R^{C \times T \times H \times W}$ of the video ($C = 1024, T = 16, H = 1, W = 1$). After this module, the ViT module is added to capture global dependency. The input of the Transformer is a standard 1D embedding. To adapt this input format, the spatio-temporal feature map is represented as a similar sequence $\mathbf{X}_t \in R^C$, $t \in \{1, 2, \cdots, T\}$ in the temporal dimension. According to the Linear project in ViT, a trainable linear mapping $\mathbf{W}$ is used to map the feature dimension from $C$ to $D$. To represent the features learned from the input sequence for classification, a trainable class token with the same shape as the obtained token is inserted after the mapping, denoted as $h_0^0 = \mathbf{X}_{\mathbf{class}}$. A learnable 1D location embedding representation $\mathbf{E}_{pos}$ is also used to facilitate classification and preserve the input feature sequence's location. Thus, the final input sequences of the Transformer are defined as follows:

$$\mathbf{h}_0 = [\mathbf{X}_{class}, \mathbf{WX}_1, \mathbf{WX}_2, \cdots, \mathbf{WX}_t]^T + \mathbf{E}_{pos} \quad \mathbf{W} \in R^{D \times C}, \mathbf{E}_{pos} \in \mathbf{R}^{(T+1) \times D} \tag{6}$$

where $\mathbf{X}_t$ denotes the $t$-th feature sequence of feature $\mathbf{X}$.

The Transformer encoder consists of a multi-head self-attention(MSA) and multi-layer perceptron(MLP) as shown in Fig 4. In addition, the LayerNorm(LN) is performed before the self-attention, and the residuals are connected. To generate the attention weights, the input sequence $h_0$ is projected into three different spaces by learning three different matrices $\mathbf{W}_q \in R^{D \times D}$, $\mathbf{W}_k \in R^{D \times D}$, and $\mathbf{W}_v \in R^{D \times D}$. Our ViT module stacks $L$ standard Transformer encode blocks. This process is formalized with the equations as follows:

$$\mathbf{f}'_l = \mathbf{MSA}(\mathbf{LN}(\mathbf{f}_{l-1})) + \mathbf{f}_{l-1}, \quad l = 1, ..., L \tag{7}$$

$$\mathbf{f}'_l = \mathbf{MLP}(\mathbf{LN}(\mathbf{f}_l)) + \mathbf{f}_l, \quad l = 1, ..., L \tag{8}$$

After the final encoding stage, an MLP head is utilized to obtain the predicted probability for the classification task. Our method is fundamentally a classification model designed for binary classification tasks. During the training process, we employ the widely-used cross-entropy loss as the loss function. The formal representation of the cross-entropy loss is as

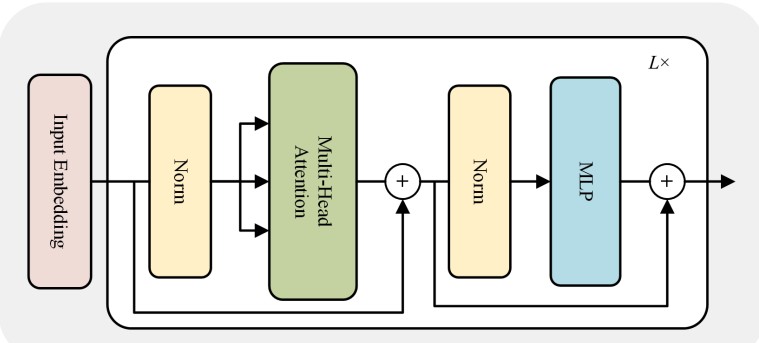

**Fig 4**. **The structure of the transformer encoder.** The ViT module contains multi-head self-attention (MSA) and multi-layer perceptron (MLP) modules to capture global dependencies in feature sequences.

follows:

$$\mathbf{L}'_{CE} = y \log \hat{y} + (1 - y) \log (1 - y), \tag{9}$$

where $\hat{y}$ is the predicted probability, and $y$ denotes the video class label, 1 for the original video, and 0 for the fake video.

## 4 Experimental evaluation

This section is organized as follows. Sect 4.1 provides a detailed description of the experiment setup. Sect 4.2 and Sect 4.3 present the results of the proposed method evaluated on intra-dataset and inter-dataset, respectively. Sect 4.4 conducts the ablation study.

### 4.1 Experimental setup

**4.1.1 Dataset.** Previous studies have extensively utilized publicly available datasets to evaluate the performance of video face forgery detection methods. In our research, we consider three widely recognized benchmark datasets: Face-Forensics++ (FF++), Celeb-DF, and DFDC.

- FF++ dataset, introduced by Rossler et al. [35], comprises 1000 original videos and 4000 forged videos. The dataset includes four sub-datasets: Face2Face (F2F), FaceSwap (FS), DeepFakes (DF), and NeuralTextures (NT). Among them, F2F generates fake videos based on real-time synchronous facial expression replacement based on 3D face modeling and image fusion; FS generates spoof videos by face swapping; DF focuses on facial replacement to generate highly realistic fake videos; NT focuses on the texture details of the characters to generate fake videos with high visual quality. Each sub-dataset corresponds to a specific forgery algorithm, showcasing various types of facial manipulations. The videos in this dataset feature complete, unobstructed faces.
- Celeb-DF, proposed by Li et al. [36], is a large-scale dataset centered around face swapping. It consists of 590 real videos collected from YouTube and 5369 fake videos generated using deepfake. The dataset encompasses a diverse range of subjects, including individuals of different ages, genders, races, and topics of interest. Notably, the synthesis process of fake videos involved an improved autoencoder and color correction techniques to enhance video quality.
- DFDC (DeepFake Detection Challenge) [37] stands as the most extensive dataset available in the field, with over 100,000 videos. The dataset includes real videos, capturing clips of 430 actors in real-life scenarios. Additionally, fake videos were generated using various faking algorithms, maintaining a ratio of approximately 1:5 in comparison to the real videos. The videos in DFDC exhibit variations in frame rates, resolutions, and approximately 5%-10% of the videos include multiple faces. For our experiments, we utilized a subset of approximately 5% of the DFDC dataset, denoted as DFDC∗. Here, DFDC∗ uses 5000 videos, including 834 real videos and 4166 fake videos.

A detailed overview of the three datasets, including their characteristics and composition, is presented in Table 1.

**4.1.2 Metrics.** Video face forgery detection is essentially a binary classification problem. In our study, we employ the following evaluation metrics to comprehensively assess the performance of our proposed method:

$$Precision = \frac{TP}{TP + FP}, \tag{10}$$

$$Recall = \frac{TP + TN}{TP + FN}, \tag{11}$$

$$Accuracy = \frac{TP}{TP + FN + TN + FP}, \tag{12}$$

**Table 1. Details of the datasets in the experiments.**

| Dataset | Volume | Intra-dataset training | Inter-dataset evaluation data (%) | Dataset partition |
|---------|--------|------------------------|-----------------------------------|-------------------|
| FF++ | 5000 | √ | 100 | 6 : 2 : 2 |
| Celeb-DF | 6229 | √ | 100 | 6 : 2 : 2 |
| DFDC* | 6400 | √ | 100 | 6 : 2 : 2 |

TP(True Positive) represents the number of samples correctly predicted as the positive class, indicating true positives. TN(True Negative) indicates the number of samples correctly predicted as the negative class, representing true negatives. FN(False Negative) refers to the number of samples incorrectly predicted as the negative class, indicating false negatives. FP(False Positive) represents the number of samples incorrectly predicted as the positive class, signifying false positives. In addition to these metrics, we also consider evaluation criteria commonly used in similar binary classification tasks to assess the performance of our model. One such metric is AUC (Area Under the Receiver Operating Characteristic Curve), which provides an overall measure of the model's discriminative ability, especially useful for evaluating imbalanced datasets. Therefore, we employ AUC as one of the evaluation metrics in some of our experiments.

**4.1.3 Experiment parameters.** Our STFANet distinguishes real and fake samples through the following steps. Firstly, the dataset is divided according to Table 1. All video data is preprocessed by slicing it into frames. MTCNN is used for face extraction, and all consecutive video frames are saved as corresponding face images. Previous studies have shown that human facial motion is typically completed within 1-2 seconds. Therefore, in our experiments, we extract features using 32 consecutive frames. The input images are resized to a fixed dimension of $224 \times 224$ during the training and testing phases, using the ResNet50 backbone network. In the ViT module, we utilize a standard transformer encoder block with self-attention heads, hidden size, and MLP size set to 8, 512, and 1024, respectively. During model training, we employ the PyTorch library with a batch size of 16, 150 epochs, an initial learning rate of 0.0004, and gradient updates using the AdamW optimizer with cosine descent. The hardware platforms used in our experiments include a CPU: Intel Xeon Platinum 8255C, memory: 64G; GPU: RTX 3090, memory: 24G. The software platform consists of PyTorch 1.7, CUDA 11.0, cuDNN 8.0.5, and a Conda environment.

## 4.2 Intra-dataset evaluation

In this section, we trained, validated, and tested the STFANet model on a single dataset to evaluate its performance. The experiments utilized preprocessed successive frames from the FF++, Celeb-DF, and DFDC* datasets as input. The experimental setup ensured that the model with the highest accuracy and lowest loss was obtained after each training iteration. The effectiveness of the model during the training process was evaluated by calculating the accuracy obtained from each training run.

**4.2.1 Experimental results.** We first evaluated the proposed method on the FF++ sub-dataset. Following the experimental settings mentioned earlier, the sub-datasets were divided into training, validation, and testing sets in a ratio of 6:2:2. The results are presented in Table 2, where it can be observed that high scores were achieved using different evaluation criteria for each sub-dataset. Our method achieved an accuracy (ACC) of over 0.9457 and an area under the curve (AUC) of 0.9850 on all sub-datasets. Learning curve plots of our model are shown in Figs 5 and 6, demonstrating almost monotonic loss and accuracy curves obtained after training the model. Since each sub-dataset utilize a single tampering technique, the introduced artifacts in the generated forged videos were relatively simple and easy to detect. These results demonstrate the effectiveness of our model in detecting videos with simple forgery patterns.

In addition, experiments were conducted on the full FF++, Celeb-DF, and DFDC* datasets to evaluate the model's ability to detect videos with complex forgery patterns. The results are presented in Table 3. Our model achieved an accuracy of over 0.9516 and an area under the curve of 0.9630 for scenarios involving multiple tampering techniques, different ages, genders, races, and other factors. The ROC curves obtained from our model experiments are shown in Fig 7.

**Table 2**. Performance of intra-dataset evaluation on sub-datasets of FF++.

| Dataset | AUC(%) | Precision | Recall | Accuracy |
|---------|--------|-----------|--------|----------|
| DF(FF++) | 99.68 | 0.9793 | 0.9595 | 0.9775 |
| FS(FF++) | 99.21 | 0.9560 | 0.9886 | 0.9666 |
| F2F(FF++) | 99.16 | 0.9638 | 0.9090 | 0.9611 |
| NT(FF++) | 98.50 | 0.9612 | 0.9636 | 0.9475 |

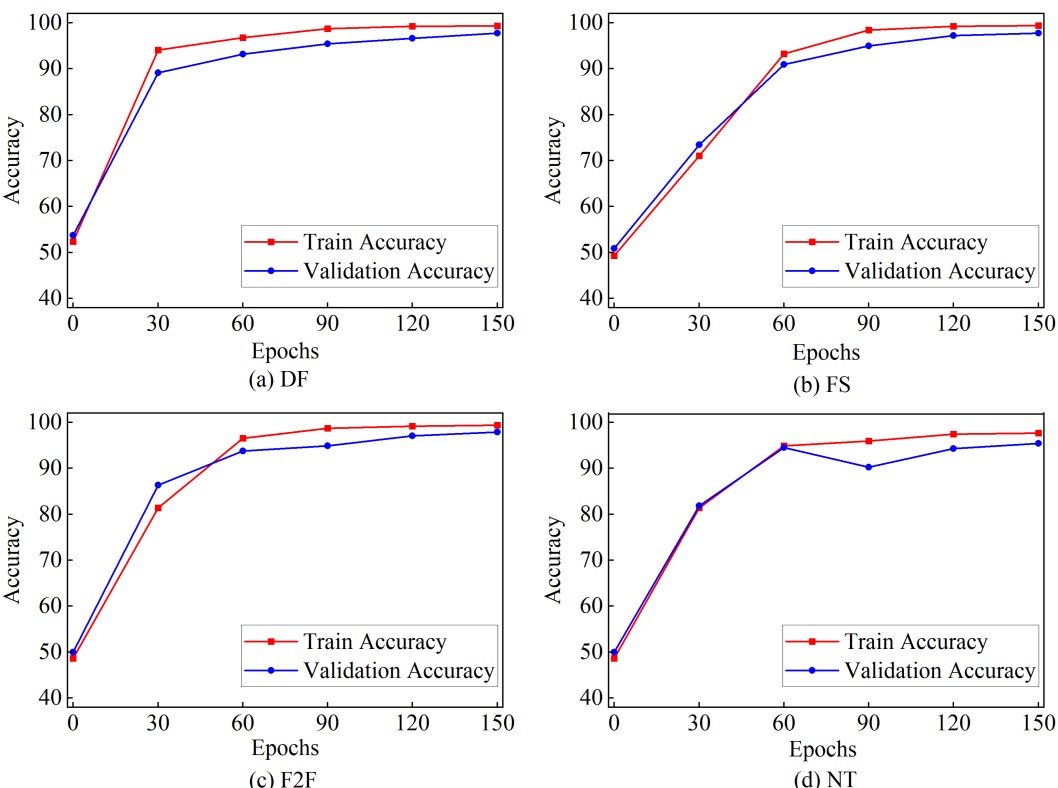

**Fig 5**. The Accuracy curve obtained by training STFANet in different sub-datasets of FF++ in intra-dataset experiments.

The results across all datasets cover almost the entire plane, indicating that our model exhibits remarkable discriminative power even for data with complex falsification patterns.

**4.2.2 Comparison with state-of-the-art methods.** The proposed method was compared with 12 existing face forgery detection using AUC as the evaluation criterion, and the results are presented in Table 4. To verify whether the observed performance differences are statistically significant, we performed a Wilcoxon signed-rank test on the AUC scores, which showed a $p$-value less than 0.05, indicating that STFANet significantly outperforms the other methods. Previous detection methods relied on exploiting the weaknesses of forgery techniques, such as MesoNet [38], which extracted low-level image features using a shallow network. However, these techniques showed degraded performance when dealing with low-quality and highly compressed images. In contrast, multi-task [3] achieved better results by multi-task learning and incorporating semi-supervised learning methods to enhance generalization on both datasets. The performance of the STFANet was also compared with LipForensics [39], M2TR [40], Multi-attentional [41], FTCN [29], Adversarial Game [42], Inspector [23], and FIC framework [24].

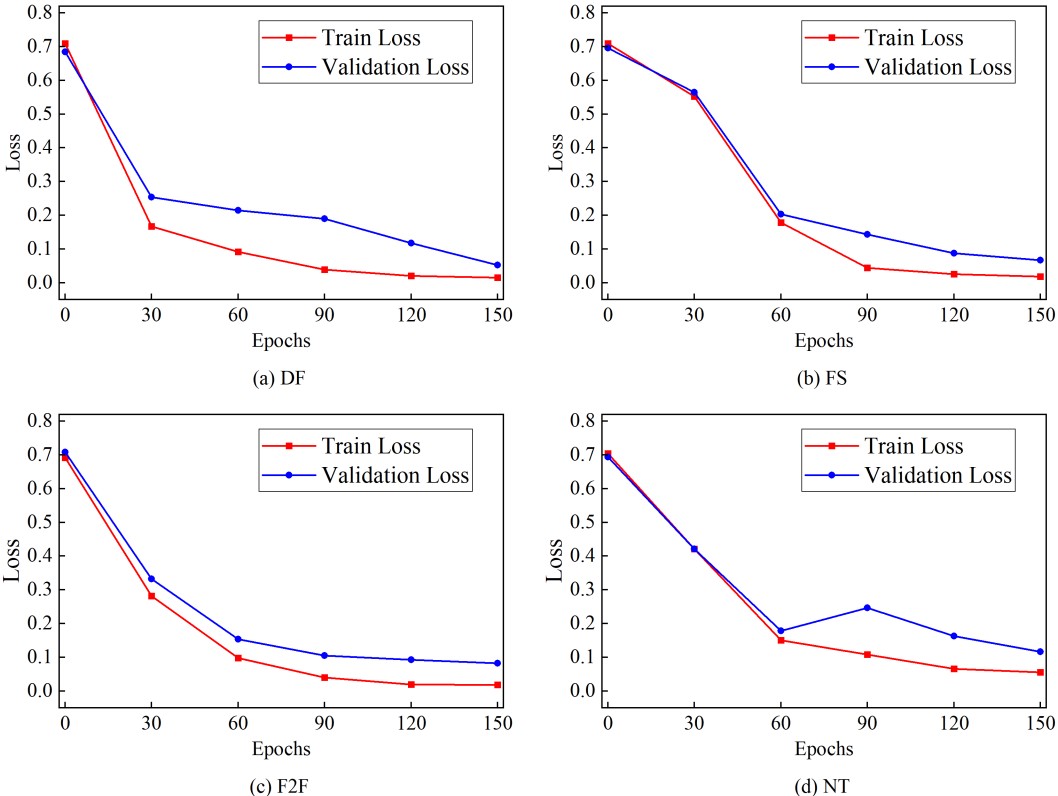

(a) DF  (b) FS

(c) F2F  (d) NT

**Fig 6**. **The Loss curve obtained by training STFANet in different sub-datasets of FF++ in intra-dataset experiments.**

**Table 3**. Performance of intra-dataset evaluation on sub-datasets of FF++.

| Dataset | AUC(%) | Precision | Recall | Accuracy |
|---------|--------|-----------|--------|----------|
| FF++ | 99.33 | 0.9507 | 0.9184 | 0.9634 |
| Celeb-DF | 98.29 | 0.9336 | 0.9016 | 0.9656 |
| DFDC* | 96.30 | 0.9031 | 0.8767 | 0.9516 |

Our method demonstrates superior performance compared to LipForensics [39] and Adversarial Game [42] on the FF++ dataset, achieving an AUC value of 0.9933 and at least 0.92% higher performance. We also observed that the video-level model performs better than the frame-level detection method, benefiting from the temporal information across frames. Additionally, our approach effectively captures the spatio-temporal inconsistent feature representation of forged videos by aggregating spatial and temporal information. The integration of the ViT module facilitates feature interactions and enables the formation of a global representation. As a result, our model achieves comparable performance to the state-of-the-art techniques FTCN [29] and Dual-level [32] on the FF++ dataset. On the more challenging Celeb-DF dataset, our STFANet surpasses existing results, such as Dual-level, by 2.89%.

## 4.3 Inter-dataset evaluation

In this section, we perform data preprocessing as described earlier. We train the model using the FF++, Celeb-DF, and DFDC⋆ datasets and assess its generalization performance through cross-validation.

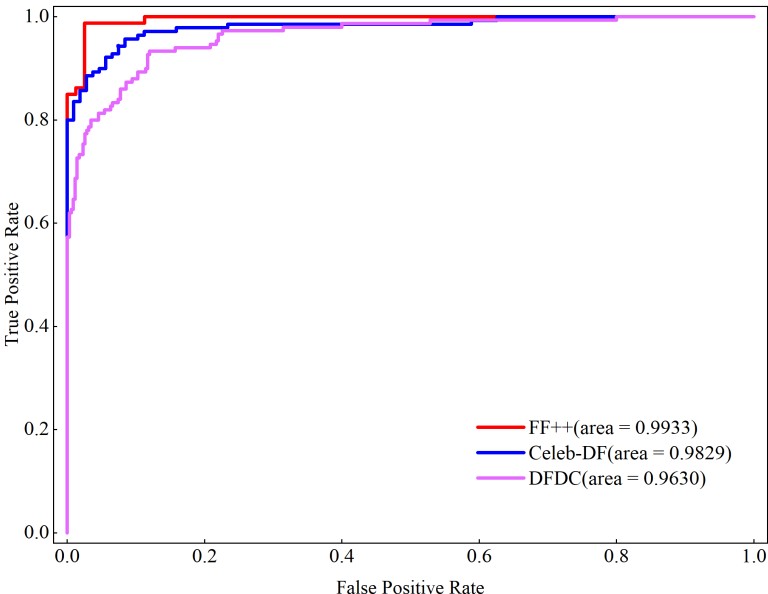

**Fig 7**. The Roc curves of FF++, Celeb-DF and DFDC with STFANet model.

**Table 4**. AUC score(%) comparisons with 12 state-of-the-art approaches on FaceForensics++ and Celeb-DF datasets.

| Method | Year | Level | FF++ | Celeb-DF |
|---|---|---|---|---|
| MesoNet [38] | 2018 | Video | 75.30 | 54.80 |
| Multi-task [43] | 2019 | Video | 80.10 | 90.50 |
| Face+Context [44] | 2021 | Frame | 75.00 | 66.00 |
| LipForensics [39] | 2021 | Video | 97.10 | 82.40 |
| Multi-attentional [41] | 2021 | Frame | 97.60 | - |
| FTCN [29] | 2021 | Video | **99.70** | 86.90 |
| Adversarial Game [42] | 2022 | Frame | 98.41 | - |
| M2TR [40] | 2022 | Video | 99.51 | 92.93 |
| Dual-level [32] | 2023 | Video | 99.20 | 95.40 |
| 3D Residual-in-Dense [31] | 2023 | Video | - | 92.93 |
| Inspector [23] | 2024 | Frame | 96.53 | 98.12 |
| FIC [24] | 2025 | Frame | 97.14 | 97.62 |
| Our method | - | Video | 99.33 | **98.29** |

**4.3.1 Experimental results.** We conducted inter-dataset tests on the four tamper types of the FF++ dataset and evaluated the generalization of our model using AUC and ACC metrics. The comparison results with Majumdar et al. [45], who used XceptionNet and ResNet50 as basic frameworks for forged video detection, are presented in Table 5. Our model outperforms existing state-of-the-art methods in most cases, but its performance on the DF, FS, and F2F sub-datasets is slightly lower than that of the state-of-the-art XceptionNet. This can be attributed to our model's focus on mining common forgery patterns across different tampering operations, leading to better generalization. Additionally, the face replacement methods (DF and FS) exhibit good generalization performance as they involve editing the entire face and share more common features. Furthermore, F2F and NT demonstrate effective generalization between each other due to the shared modules between their generative models.

To further assess the model's generalization performance, we trained it on the FF++, Celeb-DF, and DFDC* datasets and tested it on other datasets. The evaluation metrics were AUC and ACC, and the results are reported in Table 6. The

**Table 5**. Performance of inter-dataset evaluation on sub-dataset of FF++.

| Train dataset | Method | Test AUC[Test ACC](%) | | | |
|---|---|---|---|---|---|
| | | DF | FS | F2F | NT |
| DF | Our Method | **99.68[97.75]** | **72.73[67.49]** | **67.57[58.86]** | **62.61[56.69]** |
| | XceptionNet [45] | **[97.86]** | [49.93] | [51.86] | [54.07] |
| | ResNet50 [45] | - | [50.04] | [51.18] | [51.71] |
| FS | Our Method | **69.16[64.24]** | **99.21[96.66]** | **64.54[56.79]** | **66.31[55.97]** |
| | XceptionNet [45] | [52.36] | **[98.36]** | [53.61] | [48.96] |
| | ResNet50 [45] | [51.04] | - | [52.32] | [50.18] |
| F2F | Our Method | **67.36[62.06]** | **65.45[58.00]** | **99.16[96.11]** | **75.82[70.16]** |
| | XceptionNet [45] | [59.96] | [51.68] | [98.23] | [52.86] |
| | ResNet50 [45] | [55.79] | [52.18] | - | [50.79] |
| NT | Our Method | **66.53[59.16]** | **68.17[63.23]** | **71.80[66.85]** | **98.50[94.75]** |
| | XceptionNet [45] | [76.68] | [58.57] | [48.61] | [94.50] |
| | ResNet50 [45] | [74.43] | [56.39] | [48.04] | - |

**Table 6**. Performance of inter-dataset evaluation on FF++, Celeb-DF and DFDC*.

| Training dataset | Test dataset | AUC(%) |
|---|---|---|
| FF++ | FF++ | 99.33 |
| FF++ | Celeb-DF | 68.42 |
| FF++ | DFDC* | 73.14 |
| Celeb-DF | FF++ | 66.06 |
| Celeb-DF | Celeb-DF | 98.29 |
| Celeb-DF | DFDC* | 70.24 |
| DFDC* | FF++ | 58.26 |
| DFDC* | Celeb-DF | 60.29 |
| DFDC* | DFDC* | 96.30 |

experimental findings indicate that our model achieved high accuracy on known datasets and demonstrated good generalization performance on unknown datasets. However, the model trained on DFDC* exhibited lower generalization performance on FF++ with an AUC value of 0.5826. This can be attributed to the greater variety of face forgery techniques and the increased variability in the synthesis pipeline present in the DFDC* dataset compared to FF++. Fig 8 presents the confusion matrix obtained during inter-dataset testing. It indicates better generalization results between the FF++ and Celeb-DF datasets. This can be explained by the fact that both FF++ and Celeb-DF involve neural network-based face forgery methods, and our model shows reasonable generalizability to both. Moreover, our model successfully detected unseen datasets, demonstrating its accurate generalization ability.

**4.3.2 Comparison with state-of-the-art methods.** Previous researchers have proposed numerous face forgery detection methods. In this study, we compare our model with 12 state-of-the-art methods, as shown in Table 7. The comparison involves training these methods on FF++ and Celeb-DF datasets and evaluating their performance on the remaining datasets. The evaluation metric used for comparison is the AUC. Notably, the results for the DFDC* dataset, which is a subset of DFDC, are marked with an asterisk(*) to differentiate them from the other datasets.

The DSP-FWA method proposed by Li et al. [46] was a frame-level face forgery detection approach. However, its effectiveness was limited due to the lack of comprehensive datasets. The Capsule method [47] showed promising progress in video-level forgery detection on known datasets but required improvement in generalization to unknown datasets. The Two-Branch method [48] achieved excellent detection performance but would benefit from further experiments on additional datasets. Xception [36] primarily focused on image-level forgery patterns, resulting in reduced performance on unknown datasets. The multi-attention approach by Zhao et al. [41], M2TR by Wang et al. [44], and CFFs by Yu et al. [49] demonstrated improvements in detection generalization to varying degrees.

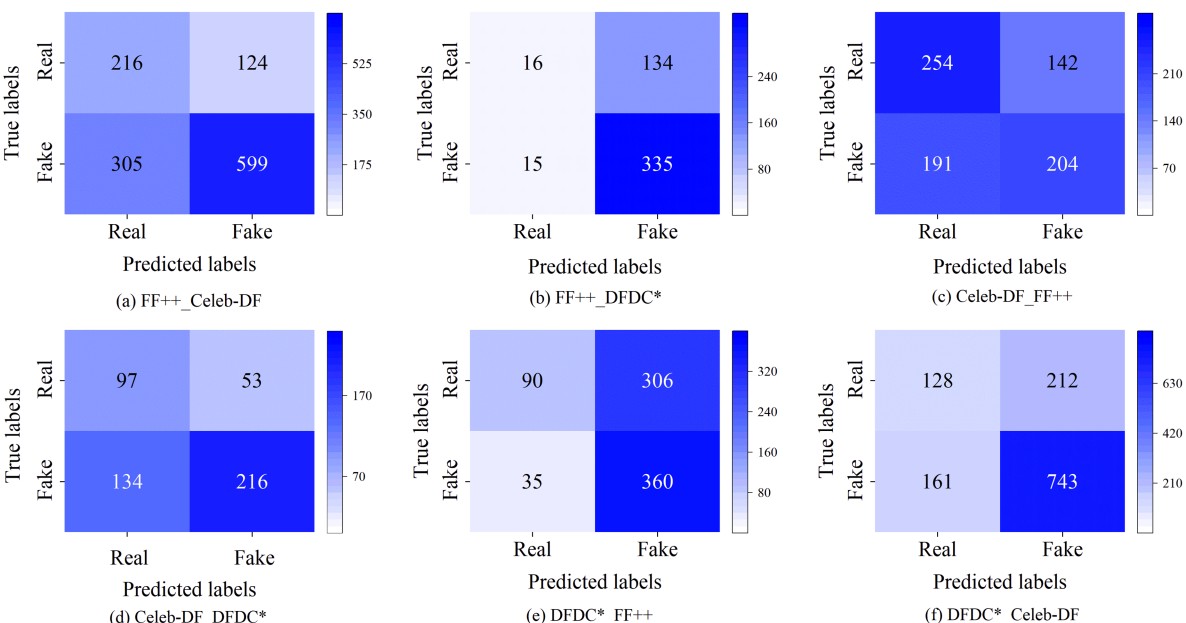

**Fig 8**. **Confusion matrix for different cross-bank experiments.** Here, $X_Y$ illustrates that $X$ is the training set and $Y$ is the test set.

**Table 7**. **Comparison of AUC scores (%) between video-based and image-based detection models on inter-dataset experiments.**

| Method | Year | Level | FF++(Train dataset) | | Celeb-DF(Train dataset) | |
|---|---|---|---|---|---|---|
| | | | Celeb-DF | DFDC | FF++ | DFDC |
| MesoNet [38] | 2018 | Video | 54.80 | - | - | 75.30 |
| DSP-FWA [46] | 2018 | Frame | 64.60 | - | - | 72.70 |
| Multi-task [43] | 2019 | Video | 54.30 | - | - | 53.30 |
| Capsule [47] | 2019 | Video | 57.50 | - | - | 53.60 |
| Two Branch [48] | 2020 | Video | 73.41 | - | - | - |
| Xception [36] | 2020 | Frame | 65.30 | - | - | - |
| Multi-attention [41] | 2021 | Frame | 67.44 | - | - | - |
| M2TR [40] | 2022 | Video | 68.20 | - | - | - |
| CFFs [49] | 2022 | Frame | 74.20 | 72.09 | - | - |
| Inspector [23] | 2024 | Frame | 66.20 | 70.5 | - | 68.67 |
| FIC [24] | 2025 | Frame | 67.20 | 69.86 | - | 67.8 |
| Our Method | - | Video | 68.42 | 73.14* | 66.06 | 70.24* |

In comparison to previous research, our STFANet achieved superior performance by addressing both spatial artifacts at the frame level and temporal information between consecutive video frames. Our method effectively captures a complete spatio-temporal feature representation of the forged video through stage-by-stage feature aggregation. While prior work primarily focused on enhancing detection accuracy within individual datasets, there is a need for more robust and effective inter-dataset evaluation. Our inter-dataset experiments demonstrated that STFANet exhibits strong generalization performance, overcoming the limitations of previous approaches.

## 4.4 Ablation study

In this section, we conducted ablation studies to evaluate the effectiveness of each component in our proposed method. The evaluations were performed on the FF++, Celeb-DF, and DFDC* datasets, and the results are summarized in Table 8 and Table 9, respectively.

**Table 8**. Ablation study of the number of feature aggregation branches.

| Dataset | Branch number | ACC | AUC |
|---|---|---|---|
| FF++ | 1 | 93.41 | 95.82 |
| | 2 | 94.93 | 97.69 |
| | 3 | **96.34** | **99.33** |
| Celeb-DF | 1 | 94.36 | 95.12 |
| | 2 | 95.96 | 96.38 |
| | 3 | **96.56** | **98.29** |
| DFDC* | 1 | 90.60 | 92.44 |
| | 2 | 93.78 | 95.58 |
| | 3 | **95.16** | **96.30** |

**Table 9**. AUC results for ablation study on vision transformer module.

| Method | FF++(train dataset) | | Celeb-DF(train dataset) | |
|---|---|---|---|---|
| | FF++ | Celeb-DF | FF++ | Celeb-DF |
| w/o vision Transformer module | 96.10 | 64.36 | 60.45 | 94.38 |
| With a vision Transformer module | 99.33 | 68.42 | 66.06 | 98.29 |

**4.4.1 Effectiveness of the number of aggregated branches.** We conducted experiments to evaluate the effectiveness of different numbers of feature aggregation branches in the detection model. The model was trained on the FF++, Celeb-DF, and DFDC* datasets, and its performance was evaluated using the AUC score. The results of the experiments are presented in Table 8. The findings indicate that performing feature aggregation only once resulted in poor detection performance, as it did not adequately represent spatio-temporal inconsistent features. However, gradually increasing the number of feature aggregation branches after the feature extraction stage improved detection performance. The best performance was achieved when three aggregation branches were used, with feature aggregation applied at all feature extraction stages. Compared to performing feature aggregation only once, increasing the number of feature aggregation branches resulted in an average improvement of 3.23% in ACC and 3.51% in AUC. These results demonstrate that increasing the number of feature aggregation branches enhances the quality of extracted features and improves detection performance.

We further utilized $t$-SNE [50] to visualize the feature representation of the spatio-temporal features after full-stage feature aggregation. The visualization is shown in Fig 9. The plot demonstrates that the spatio-temporal features are well-separated between pseudo-features and true features, indicating that our model successfully captured meaningful spatio-temporal inconsistency features through feature aggregation.

**4.4.2 Effectiveness of the vision transformer module.** We investigated the effectiveness of the ViT module in our detection model and summarized the results in Table 9. The experiments showed that when the ViT module was incorporated into the model and trained on FF++ and Celeb-DF datasets, there was an average improvement of 3.57% in AUC for intra-dataset testing. Additionally, the ViT module demonstrated a significant impact during inter-dataset evaluation, with an average performance improvement of over 4.83%. These results emphasize the valuable contribution of the ViT module in facilitating the formation of global spatio-temporal inconsistent features through feature interactions, thereby enhancing the overall performance of the model.

## 5 Conclusion

In this paper, we proposed a face forgery detection method based on spatiotemporal inconsistent features, STFANet, which uses sampling and feature aggregation at different time steps to obtain high-quality spatiotemporal features. Additionally, the STFANet model enhanced the global dependency of feature sequences by introducing the ViT module. The experiment demonstrated the excellent detection ability of STFANet on multiple datasets. It achieved AUC scores of

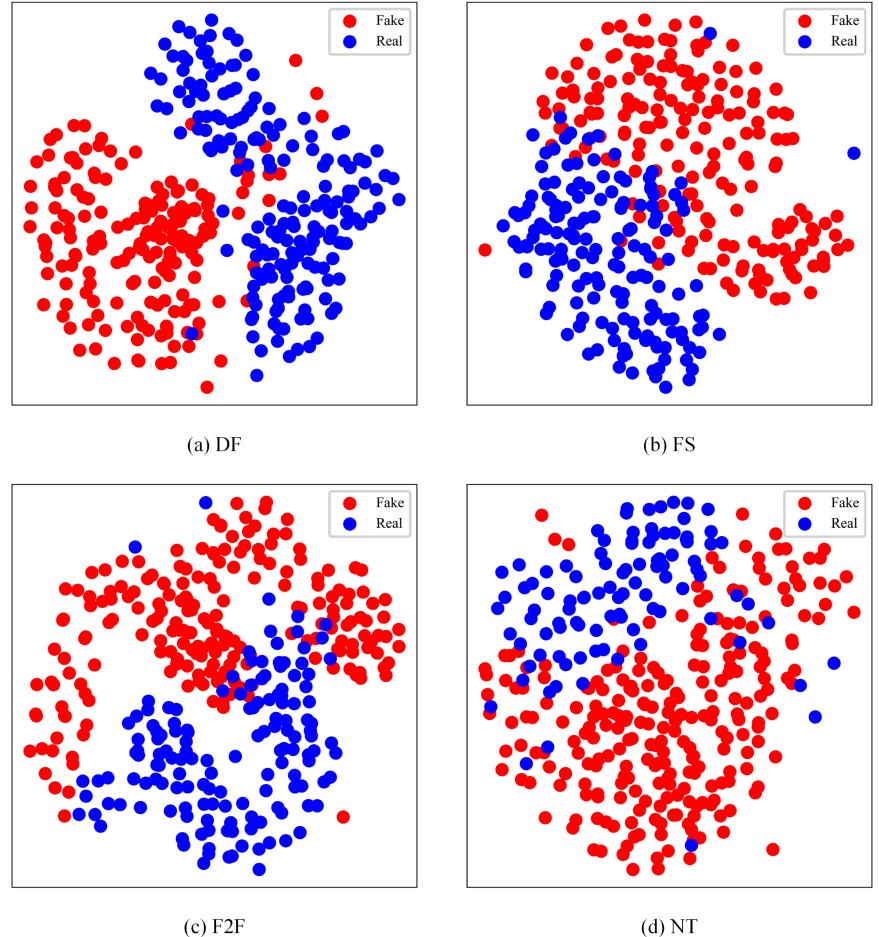

(a) DF

(b) FS

(c) F2F

(d) NT

**Fig 9**. **Representation visualization of the spatio-temporal feature on sub-datasets of FaceForensics++.** Red: Fake, Green: Real.

99.3% and 98.29% on the FF++ and Celeb-DF datasets, respectively, which are better than the state-of-the-art methods. In the cross-dataset evaluation, STFANet showed good generalization ability. When testing the Celeb-DF dataset on the FF++ training set, the AUC score reached 68.42%.

STFANet mainly uses the spatiotemporal features of video frames to detect face forgery, but face forgery may also involve information from other modalities, such as audio, text, etc. In future research, we can explore multimodal information fusion techniques, such as multi-task learning, attention mechanism, etc., to integrate information from different modalities to improve the model's ability to detect face forgery.

## Author contributions

**Conceptualization:** Guoren Yao, Gaoming Yang.

**Data curation:** Guoren Yao, Gaoming Yang.

**Formal analysis:** Guoren Yao.

**Investigation:** Xintian Liu, Lei Chen.

**Methodology:** Guoren Yao.

**Project administration:** Xintian Liu.

**Resources:** Gaoming Yang.

**Software:** Gaoming Yang, Lei Chen.

**Supervision:** Xintian Liu, Lei Chen.

**Validation:** Guoren Yao, Xintian Liu.

**Visualization:** Guoren Yao.

**Writing – original draft:** Guoren Yao.

**Writing – review & editing:** Lei Chen.

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
