## [Decision Letter · Decision Letter 0]

12 Mar 2025

PONE-D-25-07298STFANet: A Spatial and Temporal Feature Aggregation Network for Fake Face Detection in VideosPLOS ONE

Dear Dr. Yao,

Thank you for submitting your manuscript to PLOS ONE. After careful consideration, we feel that it has merit but does not fully meet PLOS ONE’s publication criteria as it currently stands. Therefore, we invite you to submit a revised version of the manuscript that addresses the points raised during the review process.

We look forward to receiving your revised manuscript.

Kind regards,

Oliver Giudice, Ph.D.

Academic Editor

PLOS ONE

5. We note that Figures 1 and 2 in your submission contain copyrighted images. All PLOS content is published under the Creative Commons Attribution License (CC BY 4.0), which means that the manuscript, images, and Supporting Information files will be freely available online, and any third party is permitted to access, download, copy, distribute, and use these materials in any way, even commercially, with proper attribution. For more information, see our copyright guidelines: http://journals.plos.org/plosone/s/licenses-and-copyright.

1. You may seek permission from the original copyright holder of Figures 1 and 2 to publish the content specifically under the CC BY 4.0 license.

Additional Editor Comments:

This paper needs some heavy rewriting in order to improve clarity. Experimental setting and design are to be better described in order to achieve reproducibility. The solution is novel but the paper needs many improvements to be considered for publication. Please refer also to reviewer 1 comments that authors need to address.

Reviewers' comments:

Reviewer's Responses to Questions

**Comments to the Author**

1. Is the manuscript technically sound, and do the data support the conclusions?

Reviewer #1: Partly

2. Has the statistical analysis been performed appropriately and rigorously?

Reviewer #1: No

3. Have the authors made all data underlying the findings in their manuscript fully available?

Reviewer #1: No

4. Is the manuscript presented in an intelligible fashion and written in standard English?

Reviewer #1: No

5. Review Comments to the Author

Reviewer #1: This study contributes to the field by introducing a novel approach for detecting fake faces in videos, differing from existing methods by utilizing the STFANet framework. Unlike conventional techniques, STFANet leverages a dual-path structure, which enhances the extraction and integration of spatial and temporal features. However, several areas require improvement to strengthen the manuscript.

The dataset usage is described, but the division into training, validation, and test sets needs more clarity. The authors should specify the exact number of videos used for each set to ensure reproducibility. Additionally, the study does not sufficiently address how the model performs against different types of manipulated videos, such as face-swapping, lip synchronization inconsistencies, or other deepfake techniques. A more detailed evaluation in this regard would enhance the comprehensiveness of the study.

While the model’s performance is compared with other state-of-the-art methods, statistical significance tests are missing. Conducting tests such as t-tests, ANOVA, or Wilcoxon signed-rank tests would help determine whether the observed performance improvements are statistically meaningful. Furthermore, the manuscript contains minor typographical errors (e.g., “experimens” should be corrected to “experiments”), and the overall readability would benefit from proofreading.

The conclusion section should more explicitly highlight the key contributions of STFANet. The advantages over other approaches should be clearly articulated, and key numerical findings (e.g., AUC scores) should be included to reinforce the model’s effectiveness. Additionally, future research directions should be elaborated, particularly regarding areas for potential improvement and further testing.

The introduction effectively establishes the significance of deepfake detection, but the originality of the proposed method needs to be emphasized more clearly. A direct statement on how STFANet specifically addresses gaps in existing research (e.g., "Our method resolves X and Y issues by...") would enhance the clarity of the contribution.

Figure descriptions are sometimes insufficient, particularly in distinguishing between Figure 5 and Figure 6. The differences between these figures should be explicitly explained so that readers can better understand the model’s components and processing flow. Additionally, Table 7 compares both video-based and image-based detection models, yet this distinction is not made clear in the table title or caption. Indicating whether each approach is image-based or video-based would improve the clarity of the comparison.

Overall, the study presents a promising approach, but these revisions are necessary to enhance its clarity, scientific rigor, and reproducibility.

6. PLOS authors have the option to publish the peer review history of their article (what does this mean?). If published, this will include your full peer review and any attached files.

Reviewer #1: No

---

## [Author Response · Author response to Decision Letter 1]

16 Apr 2025

According to the journal requirements and the reviewers' suggestions, we have answered the questions point by point and marked and modified the corresponding positions in the text. We fully respect the requirements of the journal and the opinions of the editor and reviewers, and improve and polish our manuscript to meet the publication requirements as soon as possible.

---

## [Decision Letter · Decision Letter 1]

29 Sep 2025

PONE-D-25-07298R1STFANet: A spatial and temporal feature aggregation network for fake face detection in videosPLOS ONE

Dear Dr. Yao,

Thank you for submitting your manuscript to PLOS ONE. After careful consideration, we feel that it has merit but does not fully meet PLOS ONE’s publication criteria as it currently stands. Therefore, we invite you to submit a revised version of the manuscript that addresses the points raised during the review process.

AE Comments: Update the comparison with recent works.

We look forward to receiving your revised manuscript.

Kind regards,

Sohail Saif, Ph.D

Academic Editor

PLOS ONE

Journal Requirements:

Reviewers' comments:

Reviewer's Responses to Questions

**Comments to the Author**

1. If the authors have adequately addressed your comments raised in a previous round of review and you feel that this manuscript is now acceptable for publication, you may indicate that here to bypass the “Comments to the Author” section, enter your conflict of interest statement in the “Confidential to Editor” section, and submit your "Accept" recommendation.

Reviewer #2: All comments have been addressed

Reviewer #3: All comments have been addressed

Reviewer #4: (No Response)

2. Is the manuscript technically sound, and do the data support the conclusions?

Reviewer #2: (No Response)

Reviewer #3: Yes

Reviewer #4: Yes

3. Has the statistical analysis been performed appropriately and rigorously?

Reviewer #2: (No Response)

Reviewer #3: Yes

Reviewer #4: Yes

4. Have the authors made all data underlying the findings in their manuscript fully available?

Reviewer #2: (No Response)

Reviewer #3: Yes

Reviewer #4: Yes

5. Is the manuscript presented in an intelligible fashion and written in standard English?

Reviewer #2: (No Response)

Reviewer #3: Yes

Reviewer #4: Yes

6. Review Comments to the Author

Reviewer #2: Recommendation:

Accept with minor revisions (language polishing)

The manuscript is technically sound, well-structured, and presents a meaningful contribution to the field of fake video detection. The authors have adequately addressed the previously provided reviewer comments, and the experimental results are comprehensive and convincing. I recommend acceptance of the manuscript pending minor revisions, specifically related to language polishing to enhance clarity and readability.

Reviewer #3: I have reviewed the revised submission thoroughly and am satisfied with the amendments made. The authors have addressed the points raised in the initial review comprehensively.

Reviewer #4: The authors have adequately addressed the reviewers' comments from the first round, and their responses are well-articulated and satisfactory. Besides that, I have the following comment:

The comparison with state-of-the-art methods in Table 4 lacks recent works from 2024–2025. The same applies to Table 7.

7. PLOS authors have the option to publish the peer review history of their article (what does this mean?). If published, this will include your full peer review and any attached files.

Reviewer #2: No

Reviewer #3: No

Reviewer #4: No

---

## [Author Response · Author response to Decision Letter 2]

4 Nov 2025

We would like to thank the editor and reviewers for their valuable comments and constructive suggestions. We have carefully revised the manuscript according to all the comments, and detailed point-by-point responses are provided in the uploaded response document. We sincerely hope that the revised version meets the requirements for publication.

---

## [Editor Report · Decision Letter 2]

6 Nov 2025

STFANet: A spatial and temporal feature aggregation network for fake face detection in videos

PONE-D-25-07298R2

Dear Dr. Yao,

We’re pleased to inform you that your manuscript has been judged scientifically suitable for publication and will be formally accepted for publication once it meets all outstanding technical requirements.

Kind regards,

Sohail Saif, Ph.D

Academic Editor

PLOS ONE
---

## [Editor Report · Acceptance letter]

PONE-D-25-07298R2

PLOS ONE

Dear Dr. Yao,

I'm pleased to inform you that your manuscript has been deemed suitable for publication in PLOS ONE. Congratulations! Your manuscript is now being handed over to our production team.

Kind regards,

on behalf of

Dr. Sohail Saif

Academic Editor

PLOS ONE